# Improving the efficiency of scale-up and deployment of community health workers in Mali: A geospatial analysis

**Nicholas P. Oliphant**[1,2]*, **Zeynabou Sy**[3,4], **Brehima Koné**[5], **Mohamed Berthé**[6], **Madeleine Beebe**[6], **Moussa Samake**[7], **Mamoutou Diabaté**[8], **Salimata Tounkara**[5], **Borodjan Diarra**[5], **Amadou B. Diarra**[6,9], **Cheickna H. Diawara**[6,9], **Tsvetana Yakimova**[2], **Sonia Florisse**[2], **Debra Jackson**[1,10], **Nicolas Ray**[3,4], **Tanya Doherty**[1,11]

1 University of the Western Cape, School of Public Health, Bellville, Republic of South Africa, 2 The Global Fund to Fight AIDS, Tuberculosis, and Malaria, Geneva, Switzerland, 3 Faculty of Medicine, GeoHealth group, Institute of Global Health, University of Geneva, Geneva, Switzerland, 4 Institute for Environmental Sciences, University of Geneva, Geneva, Switzerland, 5 Ministère de la Santé et du Développement Social du Mali, Direction Générale de la Santé et de l'Hygiène Publique, Bamako, Mali, 6 Ministère de la Santé et du Développement Social du Mali, L'Unité de Mise en Œuvre de Renforcement du Système de Santé, Bamako, Mali, 7 Ministère de la Santé et du Développement Social du Mali, Cellule de Planification et de Statistique Secteur Santé, Développment Social et Promotion de la Famille, Bamako, Mali, 8 Ministère de la Santé et du Développement Social du Mali, Direction National de la Santé, Bamako, Mali, 9 MUSO, Bamako, Mali, 10 London School of Hygiene and Tropical Medicine (LSHTM), Centre for Maternal Adolescent Reproductive and Child Health (MARCH), London, United Kingdom, 11 South African Medical Research Council, Health Systems Research Unit, Tygerberg, Republic of South Africa

☯ These authors contributed equally to this work.
* npoliphant@gmail.com

**Data Availability Statement:** Data are available in a public, open access repository under the Creative

## Abstract

Optimising the scale and deployment of community health workers (CHWs) is important for maximizing geographical accessibility of integrated primary health care (PHC) services. Yet little is known about approaches for doing so. We used geospatial analysis to model optimised scale-up and deployment of CHWs in Mali, to inform strategic and operational planning by the Ministry of Health and Social Development. Accessibility catchments were modelled based on travel time, accounting for barriers to movement. We compared geographic coverage of the estimated population, under-five deaths, and plasmodium falciparum (*Pf*) malaria cases across different hypothetical optimised CHW networks and identified surpluses and deficits of CHWs compared to the existing CHW network. A network of 15 843 CHW, if optimally deployed, would ensure that 77.3% of the population beyond 5 km of the CSCom (community health centre) and CSRef (referral health facility) network would be within a 30-minute walk of a CHW. The same network would cover an estimated 59.5% of U5 deaths and 58.5% of *Pf* malaria cases. As an intermediary step, an optimised network of 4 500 CHW, primarily filling deficits of CHW in the regions of Kayes, Koulikoro, Sikasso, and Ségou would ensure geographic coverage for 31.3% of the estimated population. There were no important differences in geographic coverage percentage when prioritizing CHW scale-up and deployment based on the estimated population, U5 deaths, or *Pf* malaria cases. Our geospatial analysis provides useful information to policymakers and planners in Mali for optimising the scale-up and deployment of CHW and, in turn, for maximizing the

 Supplemental appendices 1-3, all model inputs (except existing service delivery locations) and all model outputs are available from the Public Data Repository: https://doi.org/10.5281/zenodo.6551988. Health service delivery location data are only available through data-sharing agreements with the MSDS.

**Funding:** The authors received no specific funding for this work.

**Competing interests:** I have read the journal's policy and the authors of this manuscript have the following competing interests: NPO reports salary support from the Bill and Melinda Gates Foundation (BMGF) for his salary at the Global Fund to Fight AIDS, Tuberculosis, and Malaria, outside the submitted work. NPO confirms that this competing interest will not alter adherence to PLOS Global Public Health policies on sharing data and materials.

value-for-money of resources of investment in CHWs in the context of the country's health sector reform. Countries with similar interests in optimising the scale and deployment of their CHW workforce may look to Mali as an exemplar model from which to learn.

## Introduction

Achieving universal health coverage (UHC) and ensuring effective pandemic preparedness and response will require increased investment in primary health care (PHC). It will also require strengthening health systems, particularly at the primary health care level and in communities [1–4]. Community health workers (CHWs) are essential to the PHC approach as members of multidisciplinary PHC teams providing community-based PHC services tailored to population needs and preferences and serving as a trusted bridge between the health system and communities [5–8]. Investments in CHWs can be cost-effective and equity-promoting, particularly when CHWs are fairly remunerated and well-supported by the health system and communities they serve [9–14]. Investment in CHWs can also promote economic development and gender equality through fair pay in formal sector jobs, decent working conditions, opportunities for women in leadership roles, as well as social dialogue and collective bargaining [9,15–17]. However, globally there is a human resources for health (HRH) shortage, including for CHWs. The WHO estimates a deficit of 18 (range 16–19) million health workers by 2030 [18]. This deficit is exacerbated by a maldistribution of HRH, including CHWs, with the most severe effects in Africa, particularly in rural, remote, and under-served geographic areas [18–21].

As countries strive to increase sustainable financing for HRH, including for CHWs, concurrent efforts are needed to maximize the impact and efficiency of available funding through optimising the scale and deployment of HRH. Global strategies and frameworks from the WHO call for optimising the distribution of HRH and geographical accessibility to integrated PHC services [18,22,23]. Geospatial analysis using geographic information systems (GIS) can be a powerful tool in the HRH toolkit in this regard. However few countries have used geospatial analysis to optimise the scale and deployment of HRH. Previous research has focused on the use of geospatial analysis to assess the geographical accessibility of health facilities [24–26], the distribution of health facility-based HRH [27,28], and the efficiency of deployment of existing CHW networks and/or optimising the scale-up and efficiency of deployment of CHWs for subnational geographic areas [29–32] or using a Euclidean distance-based approach [33,34]. To our knowledge, only three countries have used geospatial analysis with a modelling approach based on travel-time to explore the optimization of the scale and deployment of CHWs at national scale [20–22].

In Mali, CHWs–known as *Agents de santé communautaire* or *CHWs*–have been a central part of the country's HRH at the community level since 2008. At the time of writing, the Ministry of Health and Social Development (MSDS is the French acronym) country was updating the national community health strategy in the context of a new health sector development plan and ongoing health system reform aiming to achieve UHC through primary health care [35,36]. CHWs are intended to extend equitable access to community-based primary health care services with the objective of reducing morbidity and mortality among mothers and children under-five in communities beyond 5 km of a health facility [37]. *Plasmodium falciparum* (*Pf*) malaria is a main cause of morbidity and mortality and among children under-five [37].

### Policy questions

In the context of updating the national community health strategy, the MSDS was interested in two policy questions:

1. How can we optimise scale-up and deployment of the CHWs? Given the objective to reduce morbidity and mortality among mothers and children under-five years of age, is it more efficient to deploy CHWs based on the estimated population, under-five deaths, or *Pf* (*plasmodium falciparum*) malaria cases beyond 5 km of the CSCom and CSRef network? Does one of these approaches perform best overall in terms of efficiency of deployment?

2. What percent of the population beyond 5 km of the CSCom and CSRef network can be covered by an optimised CHW network and how many CHWs are needed to do so? Comparing the existing CHW network and an optimised and scaled-up CHW network, are there deficits/surpluses of CHWs and where are the deficits/surpluses of CHWs located?

We used data from a national CHW master list and other spatial datasets in a geospatial analysis to model optimised scale-up and deployment of CHWs in Mali and inform strategic and operational planning by the MSDS. We modelled accessibility catchments based on travel time, accounting for barriers to movement, and compared geographic coverage of the estimated population, under-five deaths, and *Pf* malaria cases across hypothetical optimised networks when CHW deployment prioritised the estimated population, under-five deaths, or *Pf* malaria cases. Lastly, we compared a hypothetical optimised CHW network with the existing CHW network to identify surpluses and deficits of CHWs.

## Data and methods

### Study setting

In 2020 the health system included public, private, community, and confessional institutions organized in a decentralized, pyramidal structure with four administrative levels–a tertiary referral level, a secondary referral level, a primary referral level and a primary level–overseen by the MSDS [35]. The primary level was composed of public sector community health centres (*Centres de santé communautaire*, CSCom) and private sector health facilities staffed by nurses and–in some cases–generalist doctors providing a minimum package of primary health care services and referral/counter-referral services to/from primary referral facilities (*Centres de santé de référence*, CSRef) staffed by nurses and doctors trained on referral services (S1 Appendix 1 available via https://doi.org/10.5281/zenodo.6551988). CSCom were designed to serve the population within 5 km [37]. At the base of the primary level were paid, full-time CHWs providing community-based primary health care services, including prevention, promotion, and curative services, conducting surveillance activities, and supervising part-time community health volunteers known as *relais* [37]. The focus of our analysis was on the CHWs. The *relais* were beyond the scope of the current analysis.

According to the national community health strategy of 2016–2020, CHWs were defined as a paid, full-time CHW, recruited from, and living in the community they serve and recognized by the MSDS as meeting the minimum criteria for CHWs [37]. CHWs were allowed to provide a standard minimum package of services defined by the MSDS and implemented in the context of the national community health strategy [37]. This minimum primary health care package included prevention, promotion, and curative services [28]. This included household visits to promote reproductive, maternal, newborn, and child health and nutrition, and water and sanitation interventions; provision of family planning, integrated community case management (iCCM) of diarrhoea, pneumonia, malaria, and acute malnutrition among children

under-five, monitoring of vital events such as births and deaths, disease surveillance; participation in mass campaigns (e.g. for childhood vaccinations, distribution of seasonal malaria chemoprevention, and long-lasting insecticide-treated bednets) and supervision of the *relais* [37]. CHWs were deployed to CHW sites, i.e., villages selected by the community health association where the CHWs lived and worked and, in principle, located in rural areas beyond 5 km from a CSCom [37]. CHWs were attached to the nearest CSCom for supervision and resupply [37]. The catchment of a CHW was defined as the area within 3–4 km of the CHW site [37]. CHW sites were, in principle, the largest village within the catchment area of the CHW which also included satellite villages (i.e., villages apart from the CHW site but within the CHW catchment area and meant to be served by the CHW through outreach) [37]. The national community health strategy 2016–2020 indicated a norm of 1 CHW per 700 population in the regions of the Center and South (Kayes, Koulikoro, Mopti, Segou, Sikasso) and 1 CHW per 300–500 population in the regions of the North (Gao, Tombouctou) [37]. For our analysis, and in agreement with the MSDS, we used the ratio of 1 CHW per 700 population for the regions of the Center and South and 1 CHW per 500 for the regions of the North.

## Data

We obtained the following spatial datasets to inform our models of geographic coverage and efficiency of deployment of the CHWs: administrative boundaries (national, regional, commune) [38–40], a 2020 national georeferenced master facility list [41], a 2020 national CHW master list (CHWML) [42], digital elevation model [43], land cover [44], roads [45], official population estimates at commune level for 2020 [46], estimated population count at 100 m x 100 m resolution for 2020 [47] and travel scenarios. As of 2020, there were 3 104 working CHWs. Integrated PHC services provided by CHWs were intended to address under-five mortality, with *Pf* malaria as a major driver of curative consultations among children under-five in Mali [48]. Because the MSDS was interested to explore the efficiency of deployment of CHWs vis a vis the spatial distribution of estimated under-five deaths, in addition to the efficiency of their deployment vis a vis the estimated population, we obtained modelled estimates of the annual mean under-five mortality rate in 2017 [49] and estimated live births [50] at 5 kmx 5 km resolution to develop a raster layer for the estimated under-five deaths in 2020 at 1 kmx 1km. Similarly, because the MSDS was interested to explore the efficiency of deployment of CHWs vis a vis the spatial distribution of estimated *Pf* malaria cases, we obtained modelled estimates of the annual mean incidence of *Pf* malaria among all ages (0–99 years) in 2019 at 5 kmx 5 km resolution [51] to develop a raster layer for the estimated *Pf* malaria cases (all ages) in 2020 at 1 kmx 1km. We prepared the input datasets in the projected coordinate reference system EPSG:32629—WGS 84 / UTM zone 29N for Mali at 1 kmx 1 km resolution. We used one travel scenario, walking in dry conditions, reflecting the most relevant travel scenario for the population served by the CHWs. We prepared a travel speed table reflecting walking in dry conditions (S1 Appendix available via https://doi.org/10.5281/zenodo.6551988). We adapted travel speeds for each land cover class and road class from previous studies [20,52,53]. Travel speeds refer to the population walking in dry conditions in the direction of the CHW.

## Populations of interest

We considered three populations of interest for the first policy question:

a. the estimated population in areas beyond 5 km of a CSRef or CSCom in 2020;

b. the estimated under-five deaths in areas beyond 5 km of a CSRef or CSCom in 2020; and

c. the estimated *Pf* malaria cases in areas beyond 5 km of a CSRef or CSCom in 2020.

## Hypothetical CHW networks

We considered three hypothetical CHW networks for the first policy question (see Table 1 for definitions).

In preparation for our hypothetical scale-up CHW networks, we analysed the spatial distribution of the estimated population beyond 5 km from a CSCom or CSRef. We found that this population was predominantly located in 1 km x 1 km grid cells with an estimated population of at least 150 people. A 1 km x 1 km grid cell with an estimated 150 people is equivalent to roughly 20% of the 1 CHW to 700 population ratio for regions of the South and roughly 30% of the 1 CHW to 500 population ratio for regions of the North. We restricted potential CHW sites for our hypothetical scale-up CHW networks to 1 km x 1 km grid cells beyond 5 km of a CSCom with an estimated population of at least 150 people. This helped avoid deploying CHWs to areas with less than 20–30% of the expected CHW to population ratio, which would be an inefficient use of CHWs.

Further details on the data and methods used to derive these geographic areas are in S1 Appendix available via https://doi.org/10.5281/zenodo.6551988.

## Geographic coverage

The national community health strategy defined the catchment area of a CHW as the area within 3–4 km of the CHW site [37]. This definition ignores barriers to movement and the maximum population capacity of the CHW. To model more realistic catchment areas, we defined the catchment area of the CHWs using the concept of geographic coverage. Geographic coverage is defined as the theoretical catchment area of a health service delivery location, within a maximum travel time, accounting for the mode of transportation and the maximum population capacity of the type of health service delivery location [53]. In our

**Table 1. Definitions for the hypothetical CHW networks.**

| Hypothetical CHW network | Definition |
|---|---|
| Prioritizing population | A hypothetical CHW network deployed to prioritize geographic coverage of the estimated population in areas beyond 5 km from a CSRef or CSCom in 2020 by ordering the processing order (deployment) based on the estimated population in areas beyond 5 km from a CSRef or CSCom in 2020 within a 30-minute catchment area of a given CHW, prioritizing catchments with a higher estimated population over those with a lower estimated population. |
| Prioritizing U5 deaths | A hypothetical CHW network deployed to prioritize geographic coverage of the estimated under-five deaths in areas beyond 5 km from a CSRef or CSCom in 2020 by ordering the processing order (deployment) based on the estimated under-five deaths in areas beyond 5 km from a CSRef or CSCom in 2020 within a 30-minute catchment area of a given CHW, prioritizing catchments with a higher estimated number of under-five deaths over those with a lower estimated number of under-five deaths. |
| Prioritizing *Pf* malaria cases | A hypothetical CHW network deployed to prioritize geographic coverage of the estimated Pf malaria cases among all ages (0–99 years) in areas beyond 5 km from a CSRef or CSCom in 2020 by ordering the processing order (deployment) based on the estimated number of Pf malaria cases in areas beyond 5 km from a CSRef or CSCom in 2020 within a 30-minute catchment area of a given CHW, prioritizing catchments with a higher estimated number of Pf malaria cases over those with a lower estimated number of Pf malaria cases. |

See pages 18, 22–23 of S1 Appendix available via https://doi.org/10.5281/zenodo.6551988 for additional details on the hypothetical CHW networks.

analysis we defined geographic coverage as the estimated population (of interest) within a theoretical catchment area of the CHW network, given a 30-minute maximum travel time (walking scenario) and the maximum population capacity of the CHWs. The maximum population capacity for CHWs was based on the MSDS norms for the ratio of CHWs per population noted above. The maximum extent of an CHW catchment was therefore delimited by the maximum travel time of 30 minutes except in cases where the estimated population in the catchment exceeded the maximum population capacity. In this case, the extent of the catchment was defined by the area containing the estimated population, up to the maximum population capacity. There was no MSDS norm for the ratio of CHW per U5 deaths or *Pf* malaria cases. Assuming one CHW could cover all estimated U5 deaths or *Pf* malaria cases within their catchment regardless of population size would be unrealistic. For metrics (b) and (c) we based the number of CHWs required for the hypothetical CHW networks on the estimated number of CHW needed to cover the estimated population in each catchment using the MSDS norms above. We used the "geographic coverage" module of AccessMod 5.6.56 for all analyses [53].

## Assessing the efficiency of scale-up and deployment

We defined efficiency of deployment as the geographic coverage of the estimated population of interest achieved by a given number of CHWs, based on an adaptation of Palmer and Torgerson's definition of technical efficiency [54]. A CHW network designed to optimise the efficiency of CHW deployment maximizes geographic coverage of the population of interest with the fewest number of CHWs. This requires deploying CHWs such that each CHW maximizes the gain in geographic coverage of the population. We assessed the efficiency of deployment by comparing the gain/loss of geographic coverage for each hypothetical CHW network compared to each of the other hypothetical CHW networks, given the same number of CHWs, for each of the populations of interest.

The above analysis resulted in nine results, three results per population of interest (a-c above), and three results per hypothetical network (defined in Table 1). For each population of interest (a-c,) we compared the efficiency of deployment of CHWs across the hypothetical networks using a visual inspection of the slope of geographic coverage.

## Comparison with the existing network of CHW

For the second policy question, we used the hypothetical CHW network prioritizing the population at full scale to determine the geographic coverage of the estimated population beyond 5 km of the CSCom and CSRef networks that could be achieved, and the estimated number of CHWs needed to do so. We also estimated what could be achieved in terms of geographic coverage with the first 4 500 CHWs of the hypothetical CHW network (ranked in order of greatest contribution to geographic coverage to least contribution). We compared the hypothetical CHW network at full scale and the first 4 500 hypothetical CHWs with the existing network of CHWs to estimate deficits/surpluses of CHWs at national, regional, district, and CSCom catchment area levels. The first 4 500 CHWs of the hypothetical CHW network was used as a comparison as it presented a practical and feasible next target, given the existing network of 3 104 CHWs and anticipated levels of funding for CHWs in the near-term.

## Ethics statements

Our analysis did not include data from or about individual human participants. We did not involve patients or the public in this study.

### Ethics approval

The 2016 national georeferenced master lists of health facilities [31] and CHWs [32] were developed by the Ministry of Health and Sanitation, with support from technical and financial partners, in the context of management of the public health sector and did not require ethical approval. The protocol for secondary analysis used in this study was approved by the Ethics Committee of the University of Western Cape (Registration no: 15/7/271).

## Results

### Efficiency of deployment

A hypothetical network of 15 843 CHWs would ensure 77.4% of the estimated 2020 population beyond 5 km of a CSRef or CSCom were within a 30-minute walk of an CHW. Across the three hypothetical CHW networks, there was less than 0.6 percentage points difference in geographic coverage when prioritizing the estimated population, estimated U5 deaths, or estimated *Pf* malaria cases among all ages (0–99 years) in 2020 within a 30-minute catchment of an CHW (Table 2 and Fig 1; also see tabs "Comparison_Pop", "Comparison_U5d", and "Comparison_Cases" in S2 Appendix available via https://doi.org/10.5281/zenodo.6551988).

### Comparison with the existing network of CHW

Table 3 compares the number of CHWs needed by region and district according to a) the full hypothetical scaled-up network of CHWs prioritizing the estimated population (n = 15 843) b) the first 4 500 CHWs within the hypothetical scaled-up network of CHW (a subset of (a)) and c) the existing CHW network (n = 3 401). Column (d) provides the difference in the number of CHW between the full hypothetical network of CHW prioritizing the estimated population and the existing CHW network. Column (e) provides the difference in the number of CHWs between the first 4 500 CHW within the hypothetical network of CHW and the existing CHW network. Deficits in terms of CHWs are shown in red and surpluses are shown in blue.

Overall, there was a deficit of 12 739 CHWs between the existing CHW network (n = 3 401) and the full hypothetical CHW network (n = 15 843). The largest deficits were in the regions of Kayes, Koulikoro, Sikasso, and Ségou. Compared to the first 4 500 CHWs of the hypothetical CHW network, there was a deficit of 1 397 CHWs. For the latter comparison, the deficit was again concentrated in the regions of Kayes, Koulikoro, Sikasso, and Ségou but there were surpluses in certain districts, most notably in Commune VI of Bamako, Ansongo (region of Gao), and Bankass (region of Mopti). We provide results for the estimated deficits and surpluses of CHWs at the subdistrict level for each CSCom in Mali in S3 Appendix (available via https://doi.org/10.5281/zenodo.6551988), tab "CSCom_Comparison", located in the Public Data Repository. Fig 2 shows the 30-minute catchment area (blue) of the hypothetical CHW network prioritising geographic coverage of the estimated population in 2020. Text boxes for example CSCom indicate existing CHWs, estimated need based on the full model, estimated need based on the first 4 500 model, and deficits/surpluses comparing the existing CHW network with the models.

**Table 2. Geographic coverage of the estimated population, estimated U5 deaths, and estimated Pf malaria cases within a 30-minute catchment (walking in dry conditions) of an CHW, by three hypothetical CHW networks.**

| Hypothetical CHW network (n = 15 843) | Estimated population | Estimated U5 deaths | Estimated *Pf* malaria cases |
|---|---|---|---|
| Prioritizing population | 77.4% | 59.5% | 58.5% |
| Prioritizing U5 deaths | 77.4% | 59.8% | 58.8% |
| Prioritizing *Pf* malaria cases | 76.8% | 59.8% | 58.8% |

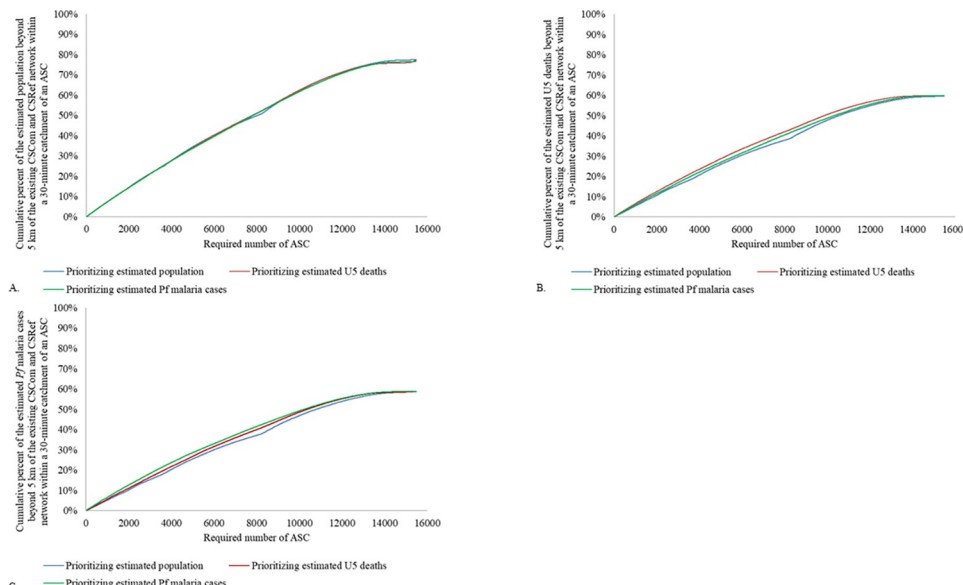

**Fig 1. Comparison of geographic coverage beyond 5 km of the existing CSCom and CSRef network according to CHW scale-up and deployment approach at 1 kmx 1 km resolution.** A) Geographic coverage of the estimated population in 2020 beyond 5 km of the existing CSCom and CSRef network covered within a 30-minute catchment area (walking) by the CHW network, according to CHW scale-up scenario; B) Geographic coverage of the estimated under-five deaths in 2020 beyond 5 km of the existing CSCom and CSRef network covered within a 30-minute catchment area (walking) by the CHW network, according to CHW scale-up scenario; C) Geographic coverage of the estimated *Pf* malaria cases in 2020 beyond 5 km of the existing CSCom and CSRef network covered within a 30-minute catchment area (walking) by the CHW network, according to CHW scale-up and deployment approach. All analyses at 1kmx 1km resolution.

## Discussion

WHO's global strategy on human resources for health, normative guidance on optimising health policy and system support for CHWs, the WHO and UNICEF operational framework for PHC, and the Working for Health Action Plan 2022–2030 call for optimising the distribution of the health and care workforce, including CHWs [5,18,23,24]. However only three previous studies have used geospatial analysis to assess the efficiency of CHW deployment at national scale using robust modelling approaches [20–22]. Champagne *et al.* compared the efficiency of various CHW deployment scenarios in terms of optimising geographic coverage of the estimated population in Haiti [22]. Oliphant *et al.* (2021) and Oliphant *et al.* (2022) compared the efficiency of CHW deployment of the existing CHW network compared to three hypothetical optimised CHW networks designed to optimise geographic coverage of the estimated population, under-five deaths, and *Pf* malaria cases, respectively, and found that the existing CHW networks were inefficiently deployed across all three targeting metrics [20,21]. However, unlike our study, these previous studies did not compare the efficiency of approaches for optimising the scale and deployment of CHWs nationally across each of these outcomes of interest [20,21]. Our study is the first to do so, providing new insight on the trade-offs (or lack thereof) between approaches and a roadmap for optimising the scale and deployment of CHWs in Mali. At the time of writing, policymakers, and planners in Mali (including authors of this study) were using our results to inform decisions on future scale-up and deployment of CHWs. As an intermediary milestone, the MSDS aims to progressively fill the gap between the existing CHW network and the first 4 500 CHWs of the optimised scale-up network that prioritized geographic coverage of the estimated population (given the

**Table 3. Estimated number of CHW needed by region and district.**

| Region | District | a) Accessmod full CHW network (n = 15 843) | b) Accessmod first 4 500 CHW | c) Existing CHWs | d) Difference c-a | e) Difference c-b |
|---|---|---|---|---|---|---|
| Kayes | Bafoulabe | 213 | 37 | 29 | -184 | -8 |
| | Diema | 318 | 98 | 24 | -294 | -74 |
| | Kayes | 417 | 167 | 45 | -372 | -122 |
| | Kenieba | 320 | 85 | 22 | -298 | -63 |
| | Kita | 453 | 62 | 73 | -380 | 11 |
| | Nioro | 254 | 85 | 9 | -245 | -76 |
| | Oussoubidiagnan | 180 | 40 | 20 | -160 | -20 |
| | Sagabari | 47 | 5 | 8 | -39 | 3 |
| | Sefeto | 75 | 43 | 3 | -72 | -40 |
| | Yelimane | 103 | 41 | 14 | -89 | -27 |
| | **Kayes Total** | **2 380** | **663** | **247** | **-2 133** | **-416** |
| Koulikoro | Banamba | 289 | 48 | 75 | -214 | 27 |
| | Dioila | 423 | 117 | 114 | -309 | -3 |
| | Fana | 339 | 110 | 106 | -233 | -4 |
| | Kalabancoro | 157 | 43 | 25 | -132 | -18 |
| | Kangaba | 169 | 66 | 53 | -116 | -13 |
| | Kati | 387 | 75 | 41 | -346 | -34 |
| | Kolokani | 538 | 68 | 66 | -472 | -2 |
| | Koulikoro | 281 | 50 | 67 | -214 | 17 |
| | Nara | 405 | 72 | 61 | -344 | -11 |
| | Ouelessebougou | 245 | 44 | 27 | -218 | -17 |
| | **Koulikoro Total** | **3 233** | **693** | **635** | **-2 598** | **-58** |
| Sikasso | Bougouni | 770 | 160 | 139 | -631 | -21 |
| | Kadiolo | 245 | 96 | 70 | -175 | -26 |
| | Kignan | 145 | 59 | 52 | -93 | -7 |
| | Kolondieba | 326 | 115 | 86 | -240 | -29 |
| | Koutiala | 589 | 156 | 95 | -494 | -61 |
| | Niena | 237 | 105 | 50 | -187 | -55 |
| | Selingue | 28 | 3 | 18 | -10 | 15 |
| | Sikasso | 445 | 117 | 87 | -358 | -30 |
| | Yanfolila | 153 | 34 | 35 | -118 | 1 |
| | Yorosso | 249 | 68 | 38 | -211 | -30 |
| | **Sikasso Total** | **3 187** | **913** | **670** | **-2 517** | **-243** |
| Ségou | Baraoueli | 285 | 89 | 37 | -248 | -52 |
| | Bla | 362 | 120 | 53 | -309 | -67 |
| | Macina | 409 | 139 | 81 | -328 | -58 |
| | Markala | 248 | 115 | 150 | -98 | 35 |
| | Niono | 411 | 198 | 93 | -318 | -105 |
| | San | 466 | 161 | 65 | -401 | -96 |
| | Ségou | 651 | 181 | 76 | -575 | -105 |
| | Tominian | 468 | 81 | 69 | -399 | -12 |
| | **Ségou Total** | **3 300** | **1 084** | **624** | **-2 676** | **-460** |

*(Continued)*

**Table 3.** (Continued)

| Region | District | a) Accessmod full CHW network (n = 15 843) | b) Accessmod first 4 500 CHW | c) Existing CHWs | d) Difference c-a | e) Difference c-b |
|---|---|---|---|---|---|---|
| Mopti | Bandiagara | 459 | 139 | 44 | -415 | -95 |
| | Bankass | 453 | 145 | 247 | -206 | 102 |
| | Djenne | 230 | 142 | 35 | -195 | -107 |
| | Douentza | 358 | 79 | 52 | -306 | -27 |
| | Koro | 642 | 288 | 45 | -597 | -243 |
| | Mopti | 308 | 114 | 31 | -277 | -83 |
| | Tenenkou | 253 | 52 | 33 | -220 | -19 |
| | Youwarou | 211 | 47 | 40 | -171 | -7 |
| | **Mopti Total** | **2 914** | **1 006** | **527** | **-2 387** | **-479** |
| Gao | Almoustrat | 9 | 4 | 0 | -9 | -4 |
| | Ansongo | 85 | 14 | 126 | 41 | 112 |
| | Bourem | 93 | 3 | 0 | -93 | -3 |
| | Gao | 72 | 10 | 20 | -52 | 10 |
| | **Gao Total** | **259** | **31** | **146** | **-113** | **115** |
| Tombouctou | Dire | 63 | 15 | 0 | -63 | -15 |
| | Goundam | 69 | 7 | 0 | -69 | -7 |
| | Gourma-rharous | 33 | 4 | 0 | -33 | -4 |
| | Niafunke | 270 | 47 | 20 | -250 | -27 |
| | Tombouctou | 59 | 38 | 0 | -59 | -38 |
| | **Tombouctou Total** | **494** | **111** | **20** | **-474** | **-91** |
| Kidal | Abeibara | 8 | 0 | 0 | -8 | 0 |
| | Kidal | 0 | 0 | 0 | 0 | 0 |
| | Tessalit | 7 | 0 | 0 | -7 | 0 |
| | Tin-essako | 0 | 0 | 0 | 0 | 0 |
| | **Kidal Total** | **15** | **0** | **0** | **-15** | **0** |
| Menaka | Anderamboukane | 11 | 0 | 0 | -11 | 0 |
| | Menaka | 7 | 0 | 10 | 3 | 10 |
| | Tidermene | 3 | 0 | 0 | -3 | 0 |
| | **Menaka Total** | **21** | **0** | **10** | **-11** | **10** |
| Taoudenit | Al-ourche | 0 | 0 | 0 | 0 | 0 |
| | Boujbeha | 0 | 0 | 0 | 0 | 0 |
| | Taoudenit | 40 | 0 | 0 | -40 | 0 |
| | **Taoudenit Total** | **40** | **0** | **0** | **-40** | **0** |
| Bamako | Commune I | 0 | 0 | 0 | 0 | 0 |
| | Commune II | 0 | 0 | 0 | 0 | 0 |
| | Commune III | 0 | 0 | 0 | 0 | 0 |
| | Commune IV | 0 | 0 | 0 | 0 | 0 |
| | Commune V | 0 | 0 | 0 | 0 | 0 |
| | Commune VI | 0 | 0 | 225 | 225 | 225 |
| | **Bamako Total** | **0** | **0** | **225** | **225** | **225** |
| **Grand Total** | | **15 843** | **4 501** | **3 104** | **-12 739** | **-1 397** |

negligible differences in efficiency between the hypothetical optimised networks). We support this approach as it is a practical and feasible near-term target given anticipated funding and it will maximize the value for money of available resources for integrated primary health care at

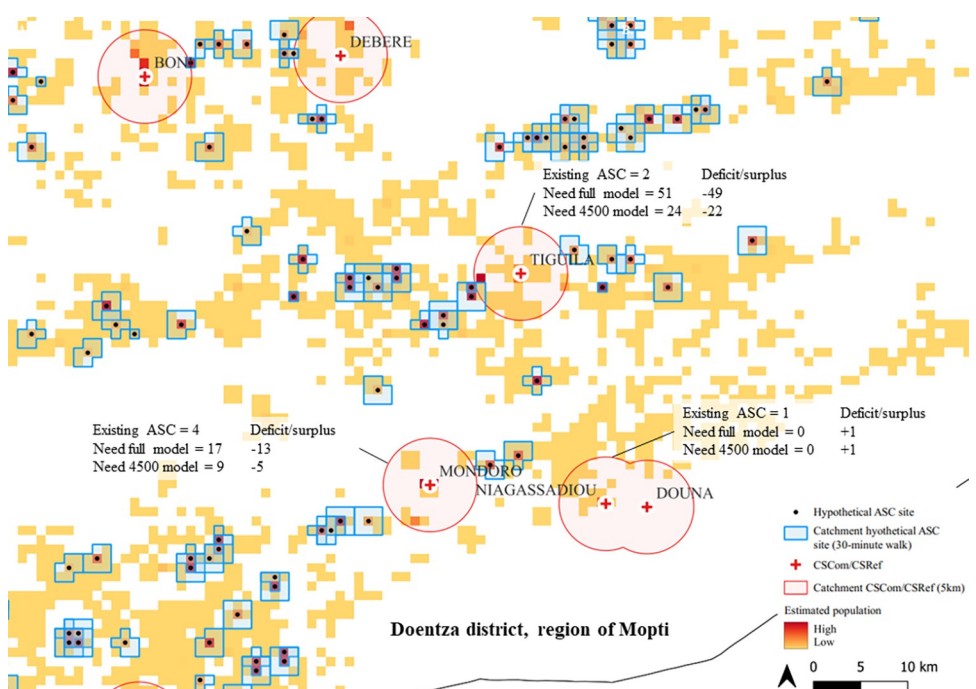

**Fig 2. Modelled 30-minute catchment areas of the hypothetical CHW network prioritizing geographic coverage of the estimated population in 2020 at 1 km x 1 km resolution.** The 30-minute catchment area (blue) of the hypothetical CHW network prioritising geographic coverage of the estimated population in 2020 based on a walking scenario and the maximum population capacity of the CHW site. Text boxes for example CSCom indicate existing CHWs, estimated need based on the full model, estimated need based on the first 4 500 model, and deficits/surpluses comparing the existing CHW network with the models. The image depicts the area around the Tiguila CSCom in the Douentza district, region of Mopti, Mali.

community level within the context of the current health system reform led by the MSDS. Recently, the Council of Ministers in Mali signed a decree officially recognizing CHWs as part of the health system. This is a remarkable milestone as it effectively lays the foundation for the possibility of domestic financing–and thereby sustainable financing–of CHWs in the future. Also of note, the WHO, at the time of writing, was planning a health labour market assessment in Mali and our results will be useful for informing that assessment as part of broader national HRH planning.

The fact that we found no important differences in geographic coverage between the approaches for scaling up and deploying CHWs has important implications for decisions on CHW deployment, as well as service integration. For example, policymakers and planners in Mali can be confident that their decision to scale up and deploy CHWs based on geographic coverage of the population adequately addresses other important concerns such as targeting the estimated burden of under-five deaths and *Pf* malaria cases. This type of analysis could be conducted in other contexts and may be particularly relevant where policymakers and planners would like to consider multiple criteria for scale-up and deployment.

While our analysis does not directly address gender equity–plans for the scale-up of CHWs and the dedicated supervisors [55] needed to effectively support the CHWs should aim to maximize gender equity of these two workforces [5]. This could be done through, for example, secondary analysis of the CHWML for the existing CHW network using a gender lens and considering affirmative action to preferentially select women during recruitment of new CHWs, following WHO guidance [5,6]. Our study also does not address CHW performance

or the optimization of the health policy and systems supports needed to maximize CHW performance [5]. These issues have been addressed previously through situational analyses and robust implementation research leading to the health sector reform and update to the national community health strategy–and will continue to be addressed in future research [36,55–58]. Planning for the scale-up of the CHW network should consider the comprehensive needs of CHWs (and their dedicated supervisors) so that they can be most effective [5,6,55,56]. For example, the participation of communities in the selection of candidates, competency-based pre-service training and accreditation, fair remuneration, dedicated supervision, equipment, job-aids and digital tools, commodities, means of transportation/funding for transportation costs for the CHWs and the dedicated supervisors for facilitated referral of patients, as well as quality improvement at CSCom and CHW levels [5,6,55–59]. Cost savings realized through the optimal deployment of additional CHWs in the future can be invested in ensuring the system components above are well-supported.

While our results point to certain CSCom and districts with an estimated surplus of CHWs according to current MSDS policy on CHW deployment, we do not recommend changing the deployment of the existing CHW network. The number of CSCom with a surplus of CHWs is small (102 CSCom) and the surplus is also small (553 CHWs). Changing the deployment of the existing CHW networks would be disruptive to the communities served, could negatively impact the trust of the affected communities in the health system, would have important negative socioeconomic impacts on the affected CHWs and their families, and would ignore the documented positive impact of CHWs in certain peri-urban areas (e.g., Yirimadio in Bamako) [58]. Instead, we support the MSDS' focus on using the results to inform future scale-up and deployment of new, additional CHWs as noted above.

As noted above 22.7% of the population remained uncovered by the hypothetical scaled-up network of CHWs. This population was in small, dispersed settlements of less than 150 people per 1 km2. To cover this population, the MSDS will need to consider the cost-benefits of different approaches e.g., 1) further expansion of the number of CHWs to such communities 2) targeting certain CHWs with motorbikes to facilitate mobile outreach by the CHWs to such communities, and 3) a combination of and 1 & 2, depending on local context.

Lastly and perhaps most importantly, to maximize the value of this kind of analysis it needs to be integrated into and updated as part of national health sector reviews and planning processes. Ideally, this kind of modelling approach would inform not only decisions on the scale-up and deployment of CHWs but also health facilities, such as the CSCom, and be considered in broader HRH and health sector strategy development and planning. As the health system expands through scaling-up CHWs and CSCom, informed by this kind of modelling, policy-makers and planners in Mali will need to periodically update the modelling as part of national reviews to account for actual health system expansion and updates to other key datasets (e.g., population). Integration of this kind of modelling into national processes as described above will be challenging. The modelling approach is data-intensive, takes time, requires a country-led approach with leadership from the MSDS, strengthening national institutional capacity, flexibility to adapt to national processes and subnational contexts, and a clear understanding of its limitations and how it can complement/be complemented by other sources of information and considerations that may be important in the decision-making process (e.g., values, political priorities). Mali has embarked on this process with this first analysis and the use of the outputs to inform national planning for the scale-up and deployment of CHWs. At the time of writing, the MSDS and development partners–including co-authors–were discussing a plan for institutional capacity building and planning the first institutional capacity building workshop to be conducted in 2022.

## Limitations

There are several important limitations of our study. First, our analysis is limited by the completeness and quality of the publicly available road and river network data. More complete and/or higher quality data on roads and rivers may be available outside the public domain. Second, estimates of the uncertainty of the estimated population counts for Mali were not available, limiting our ability to account for this source of uncertainty in measures of physical accessibility to services. Availability of this kind of data will be important for improving future modelling efforts. Third, for our targeting analysis, we resampled the modelled estimates of U5 mortality rates and *Pf* malaria incidence from 5 kmx 5km resolution to 1 kmx 1 km resolution due to lack of estimates at 1 km resolution, assuming the values for these parameters at the finer 1 kmx 1 km resolution. However, this limitation is moot given that the aim is to optimise the order of cell prioritisation (which location for a CHW should be prioritised over another), cell prioritisation is concerned with the relationship between cells (not the absolute value of cells) and the relationship between cells at 5 kmx 5 km resolution was maintained at 1 kmx 1 km resolution [20]. Third, our analysis is based on estimated travel speeds from other studies in the region [20,52,53], not empirical data from Mali or local expert knowledge, and does not account for uncertainty. Similarly, our analysis does not account for variation in walking speeds or common modes of transportation used across population groups or subnational areas. For example, pregnant women, people with illness, caregivers of ill children, the elderly population, and people with disabilities may walk slower than the general population, and predominant modes of transport may differ by geographic area or socioeconomic status. Future iterations of this analysis should attempt to address the limitations above regarding travel speeds and modes of transportation by incorporating information derived from sub-national level workshops with local experts. Fourth, our analysis did not account for the possibility of accessing health services across national boundaries, an important consideration for border communities and migrant populations. Fifth, our analysis did not account for social and economic barriers to care-seeking which may impact access to and use of health services independently of physical accessibility or through interactions with physical accessibility [60–62]. Lastly, our analysis did not consider the stockouts of equipment, supplies or commodities, quality of services and the potential for bypassing [63,64].

## Conclusion

A network of 15 843 CHWs in Mali, if optimally deployed, would ensure 77.3% of the population beyond 5 km of a CSCom or CSRef were within a 30-minute walk of a CHW. There were no important differences in geographic coverage across a range of outcomes when prioritizing scale-up based on the estimated population, estimated U5 deaths, or estimated *Pf* malaria cases. Our geospatial analysis provides useful information to policymakers and planners in Mali for optimising the scale-up and deployment of CHWs and, in turn, for maximizing the value-for-money of resources for community-based primary health care in the context of the country's health sector reform. Countries with similar interests in optimising the scale and deployment of their CHW workforce may look to Mali as an exemplar model from which to learn.

## Acknowledgments

We would like to thank the CHWs, dedicated supervisors of CHWs, other health and care workers of Mali, policymakers, and staff at the MSDS and broader Government of Mali, technical and financial partners of the MSDS, as well as civil society organizations for their contributions to the health of the population of Mali and for making this work possible through the

development of the first national master list of CHWs in Mali in 2021. #CHWsCount #PayCHWs #CountCHWs.

## Author Contributions

**Conceptualization:** Nicholas P. Oliphant, Zeynabou Sy, Brehima Koné, Mohamed Berthé, Madeleine Beebe, Moussa Samake, Mamoutou Diabaté, Salimata Tounkara, Borodjan Diarra, Amadou B. Diarra, Cheickna H. Diawara, Tsvetana Yakimova, Sonia Florisse, Debra Jackson, Nicolas Ray, Tanya Doherty.

**Data curation:** Nicholas P. Oliphant, Zeynabou Sy, Brehima Koné, Mohamed Berthé, Madeleine Beebe, Moussa Samake, Mamoutou Diabaté, Salimata Tounkara, Borodjan Diarra, Amadou B. Diarra, Cheickna H. Diawara.

**Formal analysis:** Nicholas P. Oliphant, Zeynabou Sy, Brehima Koné, Mohamed Berthé, Madeleine Beebe, Moussa Samake, Mamoutou Diabaté, Salimata Tounkara, Borodjan Diarra, Amadou B. Diarra, Cheickna H. Diawara, Tsvetana Yakimova, Sonia Florisse, Debra Jackson, Nicolas Ray, Tanya Doherty.

**Investigation:** Nicholas P. Oliphant, Zeynabou Sy, Brehima Koné, Mohamed Berthé, Madeleine Beebe, Moussa Samake, Mamoutou Diabaté, Salimata Tounkara, Borodjan Diarra, Amadou B. Diarra, Cheickna H. Diawara, Tsvetana Yakimova, Sonia Florisse, Debra Jackson, Nicolas Ray, Tanya Doherty.

**Methodology:** Nicholas P. Oliphant, Zeynabou Sy, Brehima Koné, Mohamed Berthé, Madeleine Beebe, Moussa Samake, Mamoutou Diabaté, Salimata Tounkara, Borodjan Diarra, Amadou B. Diarra, Cheickna H. Diawara, Tsvetana Yakimova, Sonia Florisse, Debra Jackson, Nicolas Ray, Tanya Doherty.

**Project administration:** Nicholas P. Oliphant, Zeynabou Sy.

**Supervision:** Debra Jackson, Nicolas Ray, Tanya Doherty.

**Validation:** Nicholas P. Oliphant, Zeynabou Sy, Brehima Koné, Mohamed Berthé, Madeleine Beebe, Moussa Samake, Mamoutou Diabaté, Salimata Tounkara, Borodjan Diarra, Amadou B. Diarra, Cheickna H. Diawara, Tsvetana Yakimova, Sonia Florisse, Debra Jackson, Nicolas Ray, Tanya Doherty.

**Visualization:** Nicholas P. Oliphant, Zeynabou Sy, Brehima Koné, Mohamed Berthé, Madeleine Beebe, Moussa Samake, Mamoutou Diabaté, Salimata Tounkara, Borodjan Diarra, Amadou B. Diarra, Cheickna H. Diawara, Tsvetana Yakimova, Sonia Florisse, Debra Jackson, Nicolas Ray, Tanya Doherty.

**Writing – original draft:** Nicholas P. Oliphant, Zeynabou Sy.

**Writing – review & editing:** Nicholas P. Oliphant, Zeynabou Sy, Brehima Koné, Mohamed Berthé, Madeleine Beebe, Moussa Samake, Mamoutou Diabaté, Salimata Tounkara, Borodjan Diarra, Amadou B. Diarra, Cheickna H. Diawara, Tsvetana Yakimova, Sonia Florisse, Debra Jackson, Nicolas Ray, Tanya Doherty.

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
