## [Decision Letter · Decision Letter 0]

15 Jun 2022

PGPH-D-22-00839

Improving the efficiency of scale-up and deployment of community health workers in Mali: a geospatial analysis

Dear Dr. Oliphant,

Thank you for submitting your manuscript to PLOS Global Public Health. After careful consideration, we feel that it has merit but does not fully meet PLOS Global Public Health’s publication criteria as it currently stands. Therefore, we invite you to submit a revised version of the manuscript that addresses the points raised during the review process.

Please submit your revised manuscript by . If you will need more time than this to complete your revisions, please reply to this message or contact the journal office at globalpubhealth@plos.org. Please include the following items when submitting your revised manuscript:

We look forward to receiving your revised manuscript.

Kind regards,

Young-Rock Hong

Academic Editor

Journal Requirements:

1. Please ensure that you refer to Fig 2 in your text as, if accepted, production will need this reference to link the reader to the figure.

2. We have noticed that you have cited Supporting Information files in your manuscript. However, there are no corresponding files uploaded to the submission. Please upload them as separate files with the item type 'Supporting Information'. Please also ensure that each Supporting Information file has a legend listed in the manuscript after the references list.

Additional Editor Comments (if provided):

The manuscript has been examined by the Editors and by external peer reviewers. We would be interested in evaluating a revised version that addresses the Comments and Editorial Requirements listed below.

Reviewers' comments:

Reviewer's Responses to Questions

**Comments to the Author**

1. Does this manuscript meet PLOS Global Public Health’s publication criteria? Is the manuscript technically sound, and do the data support the conclusions? The manuscript must describe methodologically and ethically rigorous research with conclusions that are appropriately drawn based on the data presented.

Reviewer #1: Yes

Reviewer #2: Yes

Reviewer #3: Yes

2. Has the statistical analysis been performed appropriately and rigorously?

Reviewer #1: Yes

Reviewer #2: Yes

Reviewer #3: Yes

3. Have the authors made all data underlying the findings in their manuscript fully available (please refer to the Data Availability Statement at the start of the manuscript PDF file)?

Reviewer #1: Yes

Reviewer #2: Yes

Reviewer #3: Yes

4. Is the manuscript presented in an intelligible fashion and written in standard English?

Reviewer #1: Yes

Reviewer #2: Yes

Reviewer #3: Yes

5. Review Comments to the Author

Reviewer #1: I hope it to be a fine work of the authors. The study has highlighted the importance of access and coverage of the essential health services. It has attempted to finely present the geospatial analysis on improving the population coverage and distribution of public health services to the unreached communities. It further signifies the importance of Community Health Workers (CHWs) on health service delivery.

Regarding language revisions; the author might need some proof readings. I hope it will sound better to rephrase the first sentence on the introduction as recurrently conjunction "and" has been used.

Though the national community health strategy defined the catchment area of a CHW as 3-4 km of the CHW site; what is is the rationale behind considering three populations of interest beyond 5 km of a CSRef or CSCom.

Please review the citation in the line number 17 of page 15. Please include reference for line number 22/23 for any evidence.

Besides various limitations of the study; the authors has recommended to be addressed by further researches, i hope the current modeling study could help make better policy decisions regarding the distribution of human resources for health especially at the community levels.

Reviewer #2: This is a very resourceful piece for community strategy in primary health care delivery and may be utilized to duly inform decisions around CHW deployment.

It is worth noting however, that mere physical presence of a CHW may not translate to efficiency of care delivery. This is because often these are not people with a background in health training. Their efficacy so much depends on education, day to day training, lived experience and experience working with specific populations. Perhaps, it would have been more useful touching on the level of utilization of the existing CHV network to firm up the rationale for the current study. (How well are we utilizing what we already have before reaching out for what we do not have, which we might not even afford by the way)? Would it benefit the Ministry of health more if it focused on recruiting more CHWs or managing the available CHWs? What has been the opportunity cost (what has the ministry had to forgo in order to cover/take care of the CHW shortage?

In short, if we define shortage in terms of absolute numbers/counts vs. the population then the results for the study are sound. If we look at shortage as a systemic issue that goes beyond just numbers, then we should feel the need to align investments in HRH with the current and future needs of the population and health systems. The scope of the study is well defined, therefore meets its objective. The above recommendations can open room for future research.

Reviewer #3: The paper focused on the need of scaling up human resources for health. found the paper to be analytically robust. No doubt, it has strong potential for influencing distribution policies on medical personnel particularly community health workers in Mali and elsewhere.

6. PLOS authors have the option to publish the peer review history of their article (what does this mean?). If published, this will include your full peer review and any attached files.

**Do you want your identity to be public for this peer review?** For information about this choice, including consent withdrawal, please see our Privacy Policy.

Reviewer #1: No

Reviewer #2: **Yes: **Maurine Awuor Ngoda

Reviewer #3: No

---

## [Decision Letter · Decision Letter 1]

5 Aug 2022

Improving the efficiency of scale-up and deployment of community health workers in Mali: a geospatial analysis

PGPH-D-22-00839R1

Dear Oliphant,

We are pleased to inform you that your manuscript 'Improving the efficiency of scale-up and deployment of community health workers in Mali: a geospatial analysis' has been provisionally accepted for publication in PLOS Global Public Health.

Best regards,

Rohina Joshi

Academic Editor

Reviewer Comments (if any, and for reference):

Reviewer's Responses to Questions

**Comments to the Author**

1. If the authors have adequately addressed your comments raised in a previous round of review and you feel that this manuscript is now acceptable for publication, you may indicate that here to bypass the “Comments to the Author” section, enter your conflict of interest statement in the “Confidential to Editor” section, and submit your "Accept" recommendation.

Reviewer #1: All comments have been addressed

Reviewer #2: All comments have been addressed

2. Does this manuscript meet PLOS Global Public Health’s publication criteria? Is the manuscript technically sound, and do the data support the conclusions? The manuscript must describe methodologically and ethically rigorous research with conclusions that are appropriately drawn based on the data presented.

Reviewer #1: Yes

Reviewer #2: Yes

3. Has the statistical analysis been performed appropriately and rigorously?

Reviewer #1: Yes

Reviewer #2: Yes

4. Have the authors made all data underlying the findings in their manuscript fully available (please refer to the Data Availability Statement at the start of the manuscript PDF file)?

Reviewer #1: Yes

Reviewer #2: Yes

5. Is the manuscript presented in an intelligible fashion and written in standard English?

Reviewer #1: Yes

Reviewer #2: No

6. Review Comments to the Author

Reviewer #1: Thank you for taking time to address the comments and suggestions. I hope the article will be an added value to the scientific community and people out there.

Reviewer #2: Authors of this manuscript have adequately addressed the comments earlier raised in the first instance of the review. The analysis is statistically sound and the study objective is reflected in the main findings.

7. PLOS authors have the option to publish the peer review history of their article (what does this mean?). If published, this will include your full peer review and any attached files.

**Do you want your identity to be public for this peer review?** For information about this choice, including consent withdrawal, please see our Privacy Policy.

Reviewer #1: **Yes: **Rabindra Bhandari

Reviewer #2: No
